# A New Systemic Disease Mouse Model for Glioblastoma Capable of Single-Tumour-Cell Detection

**DOI:** 10.3390/cells13020192

**Published:** 2024-01-19

**Authors:** Thomas M. B. Ware, Rodney B. Luwor, Hong-Jian Zhu

**Affiliations:** 1Department of Surgery, The Royal Melbourne Hospital, The University of Melbourne, Parkville, VIC 3050, Australia; thomas.ware@unimelb.edu.au (T.M.B.W.); rluwor@unimelb.edu.au (R.B.L.); 2Huagene Institute, Kecheng Science and Technology Park, Pukou District, Nanjing 211806, China; 3Fiona Elsey Cancer Research Institute, Ballarat, VIC 3350, Australia; 4Federation University, Ballarat, VIC 3350, Australia

**Keywords:** glioblastoma, orthotopic mouse model, single-cell detection

## Abstract

Background: Glioblastoma is characterised by extensive infiltration into the brain parenchyma, leading to inevitable tumor recurrence and therapeutic failure. Future treatments will need to target the specific biology of tumour recurrence, but our current understanding of the underlying mechanisms is limited. Significantly, there is a lack of available methods and models that are tailored to the examination of tumour recurrence. Methods: NOD-SCID mice were orthotopically implanted with luciferase-labelled donor U87MG or MU20 glioblastoma cells. Four days later, an unlabelled recipient tumor was implanted on the contralateral side. The mice were euthanised at a humane end-point and tissue and blood samples were collected for ex vivo analyses. Results: The ex vivo analyses of the firefly-labelled MU20 tumours displayed extensive invasion at the primary tumour margins, whereas the firefly-labelled U87MG tumours exhibited expansive phenotypes with no evident invasions at the tumour margins. Luciferase signals were detected in the contralateral unlabelled recipient tumours for both the U87MG and MU20 tumours compared to the non-implanted control brain. Remarkably, tumour cells were uniformly detected in all tissue samples of the supratentorial brain region compared to the control tissue, with single tumour cells detected in some tissue samples. Circulating tumour cells were also detected in the blood samples of most of the xenografted mice. Moreover, tumour cells were detected in the lungs of all of the mice, a probable event related to haematogenous dissemination. Similar results were obtained when the U87MG cells were alternatively labelled with gaussian luciferase. Conclusions: These findings describe a systemic disease model for glioblastoma which can be used to investigate recurrence biology and therapeutic efficacy towards recurrence.

## 1. Introduction

Glioblastoma is the most common malignant brain and CNS tumour with an incidence of 3.21 per 100,000 people in the US [1]. Patients currently receive an aggressive standard of care consisting of maximal surgical resection, radiotherapy and chemotherapy with temozolomide, yet overall survival remains dismal at approximately 15 months [2]. Tumour recurrence is the major cause of death, most likely originating from tumour infiltration that escapes surgical resection. Recent clinical trials have focused on assessing novel therapies against glioblastoma recurrence using a wide range of drug classes and targets, yet nearly 2000 clinical trials (including over 1000 trials in Phase II or above) have failed to provide any meaningful improvements in survival and few have led to FDA approval [3]. This is in contrast to the reported phase II and III clinical success rates of 23% and 45%, respectively, that have been observed for oncology-related drugs representing a misaligned drug development process [4]. The current inability to predict clinical responses highlights a discordance in the current approaches to clinical trial development for glioblastoma and requires a re-evaluation of therapeutic development strategies [5].

The epithelial growth factor receptor (EGFR) has been a significant target for therapeutic development in glioblastoma owing to the high prevalence of *EGFR* amplification and its important role as a key survival signalling pathway [6,7]. Initial drug development focussing on monoclonal antibodies (Cetuximab) and tyrosine kinase inhibitors (TKIs, e.g., Erlotinib and Gefinitib) has demonstrated therapeutic efficacy in preclinical mice models that measure the growth rate of primary tumours in subcutaneous and orthotopic intracranial glioblastoma implants [8,9]. Treatment with EGFR inhibitors has also improved mice survival in orthotopic xenograft models, further supporting its development into clinical trials. Despite these encouraging preclinical results, however, clinical trials have failed to recapitulate these results, with no reported benefits to overall survival in newly diagnosed or recurrent glioblastoma cases [10,11,12]. The use of EGFR inhibitors for concomitant or adjuvant therapy with the current standard of care has largely been abandoned as a clinical strategy due to these disappointing results.

Cilengitide, an αVβ3 and αVβ5 inhibitor, was developed for its potential targeting of angiogenesis and invasion, two hallmark characteristics of glioblastoma pathology. Cilengitide testing in preclinical orthotopic xenograft mice models demonstrated outstanding therapeutic efficacy, preventing primary tumour expansion beyond 1–2 mm for the duration of the experimental timeline (8–9 weeks) [13]. Phase II clinical trials, however, reported a lack of clinical efficacy in both newly diagnosed and recurrent glioblastoma [14,15]. This was further confirmed in a phase III clinical trial of MGMT methylated patients where Cilengitide failed to provide a benefit to overall survival causing the manufacture to halt development of the drug as an anti-cancer treatment [16].

Bevacizumab, a monoclonal anti-VEGF inhibitor, was once thought to be the most promising drug candidate since temozolomide and remains the most extensively clinically tested drug for glioblastoma. Early preclinical studies confirmed the specific inhibition of angiogenesis and tumour growth of primary orthotopic tumour implants, leading to an improvement in survival in mouse models [17,18]. The translation to clinical trials, however, has not been successful, providing no improvement to overall survival in newly diagnosed and recurrent glioblastoma [19,20,21,22]. Further preclinical studies looking into the failure of Bevacizumab treatment have reported that it may enhance tumour infiltration, leading to poor patient outcomes, [23] and these studies have suggested that primary tumour response may not be a predictor for clinical response.

The poor predictive value of preclinical studies testing therapeutic drugs has been observed in almost all clinical trials since the approval of temozolomide, questioning the validity of our preclinical measurements and whether they truly represent clinical targets. Current preclinical models used to predict therapeutic efficacy most often report the response of a primary tumour; clinically, however, a primary tumour is removed, and tumour response is commonly reported as the response rate or progression-free-survival, which is a measure of recurrent tumour progression. The resulting patient overall survival and cause of death are therefore attributed to recurrent tumours, which reflect unique biological mechanisms compared with primary tumour cells [24]. It is therefore expected that treatments that are effective against a primary tumour may not show the same efficacy in a recurrent setting, explaining the poor translation to clinical trials. The advancement of new therapies for glioblastoma therefore requires a deeper knowledge of the molecular mechanisms and novel targets of tumour recurrence. Advancements in this area, however, have been limited in humans due to the difficulty in obtaining quality tissue specimens and an inability to sensitively track clinical progression and identify the origin of tumour recurrence [25]. Animal models are necessary to overcome these limitations and further develop our understanding of recurrence biology. Currently, however, there are no standard mouse models available for studying tumour recurrence, with most models focusing on primary tumours. While some models have used tumour infiltration quantitation to investigate the molecular mechanisms of tumour progression, these approaches fail to capture tumour infiltration throughout the entire brain, limiting inferences regarding tumour recurrence origin [26,27].

Recent advances in tumour cell detection technology have utilised luciferase-based tumour-cell labelling for in vivo detection [28,29], including luciferase-based mouse models using human and mouse glioma cell lines [30,31,32,33]. The substrate luciferin is oxidised in the presence of a luciferase enzyme, producing a luminescence signal that is highly specific and sensitive. The luciferase system is highly reproducible and offers the additional benefit of a quantitative analysis of tumour cells. While this system has been utilised for the detection of CTCs and tumour-cell monitoring in vivo previously, single-tumour-cell detection has remained elusive [34]. Additionally, limitations in current methods often require centrifugation and cell enrichment processes, which result in the substantial loss of tumour cells and inaccurate quantification of true tumour-cell numbers. Here, we used luciferase technology and developed a systemic brain disease model of glioblastoma capable of quantitating tumour presence in mouse tissues with single-cell sensitivity.

## 2. Materials and Methods

### 2.1. Cell Cultures

The human glioblastoma cell line U87MG was obtained from the American Type Culture Collection (ATCC, Manassas, VA, USA). The human glioblastoma cell line MU20 was derived from a patient with pathologically confirmed glioblastoma at the Royal Melbourne Hospital. The use of this cell line in the laboratory was approved by the Melbourne Health Human Research and Ethics Committee (HREC 2012.219). The cells were maintained in Dulbecco’s Modified Eagle’s Medium (Thermo Fisher Scientific, Waltham MA, USA) containing 10% foetal bovine serum (Thermo Fisher Scientific), 2 mM glutamine, 100 U/mL penicillin and 100 μL/mL streptomycin (Invitrogen, Carlsbad, CA, USA). The cells were incubated in a humidified atmosphere of 10% CO_2_ at 37 °C.

### 2.2. Stable Transfection of the Firefly and Gaussia Luciferase

The U87MG and MU20 cells were stably transfected with pGL4.51[luc2/CMV/Neo] (Promega, Madison, WI, USA), and the U87MG cells were stably transfected with a pCMV-Gluc-KDEL (Prolume Ltd., Pinetop, AZ, USA) vector using Fugene^®^ HD (Promega). The cells underwent selection with 2 mg/mL G418 (Roche, Basel, Switzerland), and stable clonal colonies were obtained.

### 2.3. Animal Experiments

The animal experimentation was approved by the University of Melbourne Animal Ethics Committee (Project ID 10012). Six-to-eight-week-old female NOD-SCID mice (Animal Resource Centre, Perth, WA, Australia) were used. For the intracranial implantation, the mice were anaesthetised with 4% isoflurane and maintained at 2% isoflurane for the procedure. The mice were shaved and positioned on a stereotactic frame (Parkland Scientific, Inc., Coral Springs FL, USA). A scalp incision was then made, and a small burr hole < 1 mm diameter was fashioned and 5 × 10^4^ or 2.5 × 10^5^ firefly-labelled cells were implanted in 5 µL (PBS) using a 27-gauge needle at a flow rate of 1 µL/min. The needle was retracted, and the hole was sealed using sterile bone wax. The mice were monitored for recovery, and 4 days later, a non-labelled contralateral tumour was implanted using the same procedure. This model mimicked several other contralateral models we and others have used to investigate tumour cell infiltration/invasion or therapeutic interventions [35,36,37]. Following the experimental end-point (based on daily monitoring and the condition of the mice and ranging from 13–39 days after implantation of the cells), the mice were euthanised, and blood samples were collected and their organs were snap-frozen. For cerebral blood collection, the jugular vein was ligated above the clavicle and the vein was incised.

### 2.4. Ex Vivo Luciferase Analysis

Mouse brain and lung tissues and mouse whole blood samples were collected from a control (mice that were not injected with tumour cells) and tumour-inoculated mice. The extracted tissues were then cut into ~25 mg pieces and homogenised in a cell culture lysis reagent (Promega). The tissue lysates were maintained at 4 °C for 30 mins with agitation and either transferred directly to an opaque 96-well plate or “spiked” with a titration of U87MG-Fluc or MU20-Fluc cells before being added to the opaque 96-well plate. Luciferase readings were performed using a luciferase assay kit (Promega) and a luminometer (Promega). The luciferase activity was presented as relative luciferase units (R.L.U).

### 2.5. Statistical Analyses

All statistical analyses were performed in Graphpad Prism 6 (GraphPad Software Inc, Boston, MA, USA; v6.01). Single cell quantitation was determined using a best-fit linear regression analysis, and cell numbers were extrapolated for positive luciferase samples. An unpaired student’s *t* test was used for comparisons between the groups.

## 3. Results

Detection of single glioblastoma cells in vitro: While glioblastoma is known to be a highly invasive brain tumour, most studies have failed to consider the consequences of tumour infiltration when developing new treatments in a preclinical setting. We therefore sought to develop a model capable of accurately detecting all tumour cells present within all sections of a mouse brain. To achieve the required sensitivity, we used luciferase labelling, which can provide a reproducible luminescence signal with minimal background, and it is capable of single-cell detection.

The U87MG cells stably transfected with the firefly luciferase construct pGL4.51 were added to the mouse brain tissues ex vivo in serial dilutions to determine their sensitivity down to single tumour cells. The regression analysis reported a strong linear relationship (R^2^ = 0.99), with minimal variation in the luciferase signal. The single tumour cells produced a signal of 464 relative luciferase units (R.L.U) per cell in 5 mg of brain tissue, which was clearly identifiable above the background signal of 19 R.L.U for the control brain tissue (approximately twenty-five-fold higher; Figure 1A), providing sensitive and specific detection of single tumour cells in the mouse brain tissues.

Recently, the presence of circulating tumour cells (CTCs) in blood has been widely reported in the glioblastoma setting, though the reliable detection of CTCs remains difficult [38,39]. We therefore serially titrated our U87MG-Fluc cells in mouse blood and lung tissue ex vivo for evidence of potential CTCs. A single-cell signal of 58 R.L.U in 5 µL of blood and 362 R.L.U in 5 mg of lung tissue (R^2^ = 0.99) was determined (Figure 1B,C). However, the detection of luciferase activity in the absence of tumour cells provided a background reading of approximately of 12 R.L.U for blood and lung tissue, which was a negligible signal compared to the single-tumour-cell activity (approximately a five-fold increase for the blood and a thirty-fold increase for the lung tissue). These results therefore demonstrated a sensitive and reproducible method for single-tumour-cell detection in mouse blood and lung tissue. We next determined that these results were repeatable using another stably transfected glioblastoma cell line (MU20). Similarly, we determined the single-cell signal within brain, blood, and lung samples from the MU20-Fluc (Appendix AA–C).

Detection of Single Glioblastoma Cells In vivo: Having confirmed sensitive single-tumour-cell detection in vitro, we next evaluated whether we could detect single tumour cells in vivo by using a contralateral orthotopic glioblastoma mouse model. This model has been used in other cancer settings to evaluate tumour cell infiltration or the efficacy of therapeutics [35,36]. The U87MG-Fluc and MU20-Fluc donor glioblastoma tumour cells were implanted into the basal ganglia of one hemisphere, while unlabelled recipient glioblastoma tumour cells were implanted in the contralateral hemisphere 4 days later (Figure 2A). Peripheral brain samples were collected from the frontal lobe and caudal side of the parietal-temporal/occipital lobe distal to the primary tumours (Figure 2B). The mice implanted with MU20 tumours had a shorter median survival time of 28 days compared to 36 days for the U87MG-implanted tumours (*p* = 0.03) (Figure 2C), and they displayed increased cellularity and invasive patterns (Figure 2D–G). The MU20-Fluc-labelled tumours displayed an aggressive invasive margin, with tumour clusters leading the invasive front. In contrast, the U87MG-Fluc tumour cells exhibited contained and expansive growth of the primary tumours, with no identifiable invasive margin present.

To characterise the extent of tumour infiltration around the brain, we first examined the luciferase activity in our labelled donor and unlabelled recipient U87MG tumour explants. We detected positive luciferase signals in the unlabelled recipient tumour explant compared to the negative control non-implanted mouse brain tissue (*p* = 0.01) (Figure 3A). Furthermore, analysis of the area surrounding where the unlabelled tumour cells were injected (the designated peripheral brain tissue) was positive for luciferase activity in comparison to the corresponding non-implanted control mouse brain (*p* = 0.002) (Figure 3B). Luciferase activity was also detected in the blood and lung samples of the mice (blood, *p* = 0.04; lung tissue, *p* < 0.001). Intriguingly, while luciferase activity was detected in the lungs of all the mice, luciferase activity was only detected in the blood of three of the four mice (Appendix AA). Luciferase signals were also detected in other sampled mice organs (Appendix AA). Quantitation of the U87MG-Fluc tumours using the standard curve from Figure 1 determined a mean of 5325 cells per 5 mg of brain tissue (range: 531–11,116 cells/5 mg) in the firefly-labelled donor explant and 3.9 cells per 5 mg of brain tissue (range: 0.8–10.3 cells/5 mg) in the unlabelled recipient explant (Figure 3C). The tumour-cell quantitation also demonstrated the presence of low tumour-cell numbers in all peripheral brain samples (mean: 1.3 cells/5 mg and range: 0.2–4.2 cells/5 mg) and lung tissue samples (mean: 0.4 cells/5 mg and range: 0.1–0.6 cells/5 mg). Since not all mice were positive for luciferase activity and they were euthanised at different times, only the mice with positive blood samples were quantitated to determine the relative CTC number when present. In these jugular blood samples, a mean of 3.5 cells per 5 µL of blood (range: 1.0–6.6 cells/5 µL) was detected (Figure 3D). These results characterised a systemic dissemination of glioblastoma cells across both hemispheres of the brain and in the blood.

We next set out to characterise the infiltration of the MU20-Fluc-implanted mice. Similar to our results from the U87MG-Fluc-implanted mice, luciferase activity was detected in the labelled donor and unlabelled recipient tumours as well as all peripheral brain and lung tissue samples collected (unlabelled recipient tumour, *p* = 0.009; peripheral brain, *p* = 0.04; and lung tissue, *p* < 0.001) (Figure 4A,B). CTCs were also detected in the blood samples of three of the four mice (*p* = 0.03). Interestingly, most mouse organ tissues tested were positive for luciferase signals (Appendix AB). Quantitation of the tumour cell numbers in the labelled donor tumours determined a mean of 1,372,968 cells per 5 mg of brain tissue (range: 23,889–3,182,974 cells/5 mg) and 3898 cells per 5 mg of brain tissue (range: 244–8121 cells/5 mg) in the unlabelled recipient tumour (Figure 4C). Tumour cells were also present in all peripheral brain samples (mean: 695.2 cells/5 mg, range: 33.9–2399 cells/5 mg), but there was a comparatively reduced tumour presence in the blood (mean: 6.8 cells/5 µL, range: 1.0–16.6 cells/5 µL; positive samples only) and lung tissue samples (mean: 81.0 cells/5 mg, range: 9.9–144.4 cells/5 mg) (Figure 4D).

To ensure that the observed phenotype was not the result of firefly luciferase labelling, we generated a non-secreting gaussia luciferase-labelled U87MG cell line. Single-cell sensitivity was confirmed in vitro as described earlier (R^2^ = 0.99). The U87MG-Fluc-labelled donor cells were intracranially implanted into the basal ganglia of the mice, and the unlabelled recipient U87MG cells were implanted contralaterally 4 days later. Given that our previous model presented with detection of the U87MG tumour cells within the peripheral brain tissue and blood of the mice as early as 13 days post-implantation, we euthanised all mice at 16 days to test the consistency of our model. Consistent with our previous results, all mice tissue samples were positive for gaussian luciferase activity (unlabelled recipient tumour, *p* = 0.03; peripheral brain, *p* < 0.001; and lung tissue, *p* = 0.04) (Figure 5A,B), though we did not detect any signal within the collected blood samples (*p* = 0.31). A quantitative analysis of the tumour cell numbers demonstrated an average (mean ± SEM) of 734.0 ± 248.6 cells per 5 mg of brain tissue within the labelled donor explant and 8.1 ± 4.8 cells per 5 mg of brain tissue within the unlabelled recipient explant (Figure 5C). In the peripheral brain samples, an average of 1.4 ± 0.15 cells per 5 mg of brain tissue was detected, which was lower than the presence of tumour cells within the lung tissues (7.5 ± 2.4 cells/5 mg) (Figure 5D). These results further confirmed our findings of a systemic infiltration of tumour cells within the brain and the likely presence of CTCs, though this was not detected in our U87MG-Fluc-implanted mice.

## 4. Discussion

Infiltration of the brain parenchyma is thought to be the cause of tumour recurrence and patient mortality in glioblastoma [40]. The extensive dissemination throughout the brain evades primary surgical resection and remains incurable with the current standard of care [2]. While recurrent clinical trials can test the specificity of a drug for recurrent tumour cells, there is often limited existing preclinical evidence for efficacy in recurrent tumours, and this has resulted in expensive and long clinical trials that have ultimately yielded unsuccessful outcomes [3,41]. Better models are needed to predict treatment response in recurrent glioblastoma and, importantly, to establish the mechanisms regulating tumour invasion and recurrence.

In this study, we developed a novel quantitative model of glioblastoma tumour infiltration. We reported the presence of glioblastoma tumour cells moving away from the labelled donor tumour to the unlabelled recipient tumour in both U87MG and MU20 xenograft cell lines. Remarkably, the detection of tumour cells within the unlabelled recipient tumour was observed after only 13 days in the U87MG xenograft model, suggesting highly motile tumour dissemination beginning in the early stages of tumour development. The labelled tumour cell numbers increased in the unlabelled recipient tumour over time in both xenograft models, possibly due to expansion of the labelled cells in the recipient tumours and/or continual infiltration from the labelled donor tumours. Interestingly, tumour cell infiltration was detected extensively throughout the distal regions of the supratentorial brain tissues. A similar finding was reported by Sahm et al. (2012), who described the presence of single glioblastoma tumour cells in peripheral and inconspicuous regions of the human brain [42]. The authors stated that single tumour cells were located in virtually most areas of the brain, consistent with a systemic disease of the brain. The validity of the conclusion for primary glioblastoma, however, is questioned by the use of the IDH1 mutation as a marker for tumour cells since IDH1 mutations in glioblastoma confer to a genetically distinct and infrequent subset of glioblastoma, with better survival outcomes being predicted [43,44]. Our current study has confirmed and expanded on the extensive pattern of infiltration across the supratentorial brain in primary glioblastoma, further supporting the concept of a systemic brain disease.

These findings, however, are unexpected for the U87MG xenograft model, which is commonly criticised for not recapitulating the invasive tumour margin seen clinically in glioblastoma [45,46]. The U87MG xenograft in our study failed to demonstrate invasion at the primary tumour margin, yet it still progressed to the whole brain. Until recently, most brain tumour researchers and clinicians believed that haematogenous dissemination was not common in glioblastoma due to the rare incidence of clinical metastases [47,48]. With advances in CTC detection technology, blood-borne glioblastoma cells have been isolated and gradually accepted as a characteristic of glioblastoma pathology, though the clinical relevance and regulation of these CTCs remain unclear [49,50]. Recently, it was reported that glioblastoma CTCs self-seed to a primary tumour in vivo, establishing a potential mechanism for hematogenous dissemination within the brain [51]. Here, we detected both viable CTCs and lung micrometastases in both the U87MG and the MU20 xenograft models, supporting a functional pathway for haematogenous dissemination. Although lung metastases have not been reported in glioblastoma patients, we cannot rule out that micro-metastases may occur, though at such low cell numbers that they are undetectable or that lung screening is simply not performed in the glioblastoma setting. CTCs may offer a novel mechanism for tumour dissemination throughout the brain and explain the extensive dissemination of the U87MG xenograft model, despite its limited invasive capabilities. It will be of interest to further elucidate the molecular mechanism of how these CTCs are generated in glioblastoma and its implications for tumour recurrence.

To advance future therapeutic strategies for glioblastoma, it will be important to recognise the biological development of recurrence. There remains, however, much debate about the role of progressive genetic mutations in the formation of recurrence. Körber et al. (2019) provided the most comprehensive overview of recurrence development using deep whole-genome-sequencing on primary and recurrent human glioblastoma tumours. They reported that relapsed tumours did not acquire stereotypical patterns of mutations in recurrence and progressed from existing oligoclonal populations from the primary tumours [52]. This finding suggests that glioblastoma develops early and has acquired its driver mutations by the time of diagnosis. Our novel mouse model extended on this finding, using the existing genetic mutations of our xenograft cell lines to investigate tumour progression. Expanding on our findings, we propose that glioblastoma recurrence is preordained at the time of diagnosis whereby glioblastoma tumour cells have already infiltrated systemically throughout the brain. It is possible that this extensive infiltration of the brain may lead to residual cells that, given time, are capable of forming tumour recurrence. In 1928, Walter Dandy famously demonstrated that radical hemispherectomy of a diseased brain was not a cure, with tumour recurrence developing in the contralateral hemisphere [40]. These early experiments provided evidence of widespread dissemination of glioblastoma throughout the brain that is not captured in current animal models. Specifically, limitations in tumour detection technology prevent close examination of the inconspicuous locations of the brain where residual tumour cells may reside and form eventual recurrence [40]. This provides significant implications for future treatment directions and presents the possibility of perpetual recurrence should residual tumour cells remain following treatment. Future research directions will require a comprehensive understanding of the distinct recurrent cell entities that exist at the time of diagnosis, and this challenges our current focus on primary tumour biology.

The ability to measure tumour infiltration in vivo with the use of this model may provide a significant advancement over existing preclinical models that have failed to predict therapeutic efficacy in a clinic. This is critical as our model could offer an improved in vivo method to test the efficacy of novel and repurposed therapeutics in a recurrent setting. However, one limitation of our model was the inability to evaluate any anti-tumour immune response promoted by potential therapeutics that could elicit these immune-based responses. In addition, despite obtaining clear, significant outcomes in our animal model, greater numbers of mice will need to be used to validate other similar models in glioblastoma and other cancer settings. Nonetheless, our standardised mouse model allowed for a highly sensitive comparative analysis of tumour infiltration, representing the major failure of current therapeutic development and presenting distinct biological properties to a primary tumour.

In summary, our single-tumour-cell detection method has uncovered novel characteristics of glioblastoma progression that emphasise glioblastoma as a systemic disease of the brain. This model improves on previous methods of tumour cell detection and quantification, which have been limited by sensitivity or specificity. Our method overcomes these limitations, providing highly sensitive single-tumour-cell quantitation capable of detecting viable tumour cells with no cell loss. The sensitive nature of our luciferase system is ideal for both short- and long-term studies on tumour cell dissemination. Additionally, due to the small blood volume required for analysis, longitudinal monitoring of tumour cells can be performed to investigate the temporal relationship between early tumour cell dissemination and potential recurrence. Our method can provide important preclinical information regarding the mechanisms of glioblastoma tumour progression and recurrence under controlled conditions.

## 5. Conclusions

Through characterization of a novel glioblastoma infiltrative mouse model, we identified glioblastoma as a systemic brain disease with single tumour cells infiltrating all regions of the supratentorial brain. Further investigation and use of this mouse model can help advance our understanding of glioblastoma progression and provide new strategic research directions for the development of new predictive biomarkers and targeted therapies against recurrence. Our model also offers an improved in vivo screening tool for therapeutic interventions in glioblastoma infiltration and recurrence.

## Figures and Tables

**Figure 1 cells-13-00192-f001:**
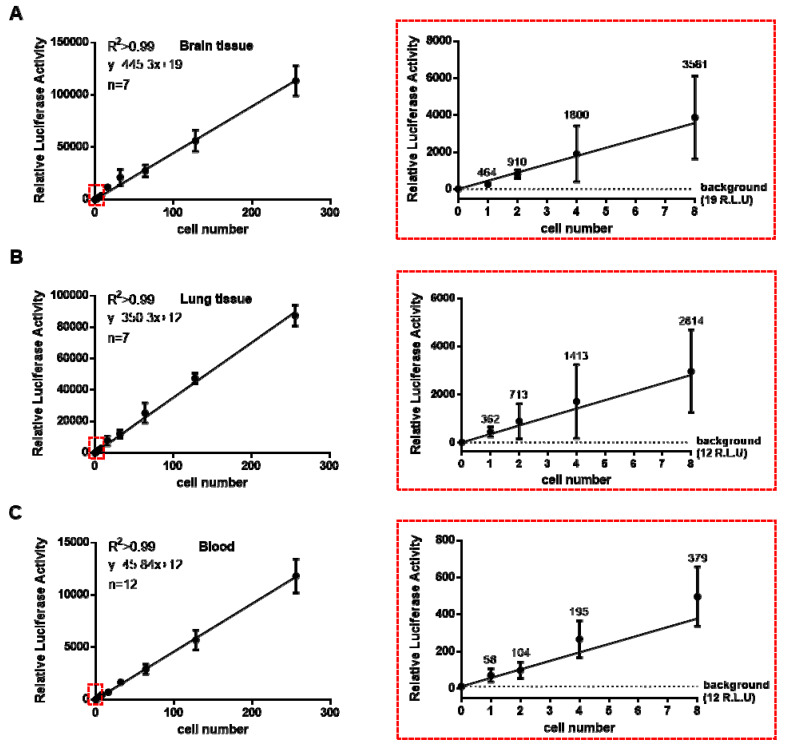
U87MG-Fluc single-tumour-cell quantitation in vitro. The U87MG-Fluc cells were serially diluted and spiked into 5 mg of brain tissue (**A**), 5 mg of lung tissue (**B**) or 5 µL of blood (**C**). The cells were lysed and the relative luciferase activity (R.L.U) was measured (**left panels**). Single-cell quantitation was determined by a best-fit linear regression analysis and calculated at approximately 464 R.L.U, 362 R.L.U and 58 R.L.U for the brain, lung and blood samples, respectively ((**right panels**) (zoomed-in red box)). The luciferase background was 19 R.L.U for the brain tissue and 12 R.L.U for both the lung tissue and blood without tumour cells. The data shown are means ± SEMs, with n = 7 for the brain and lung tissues and n = 12 for the blood.

**Figure 2 cells-13-00192-f002:**
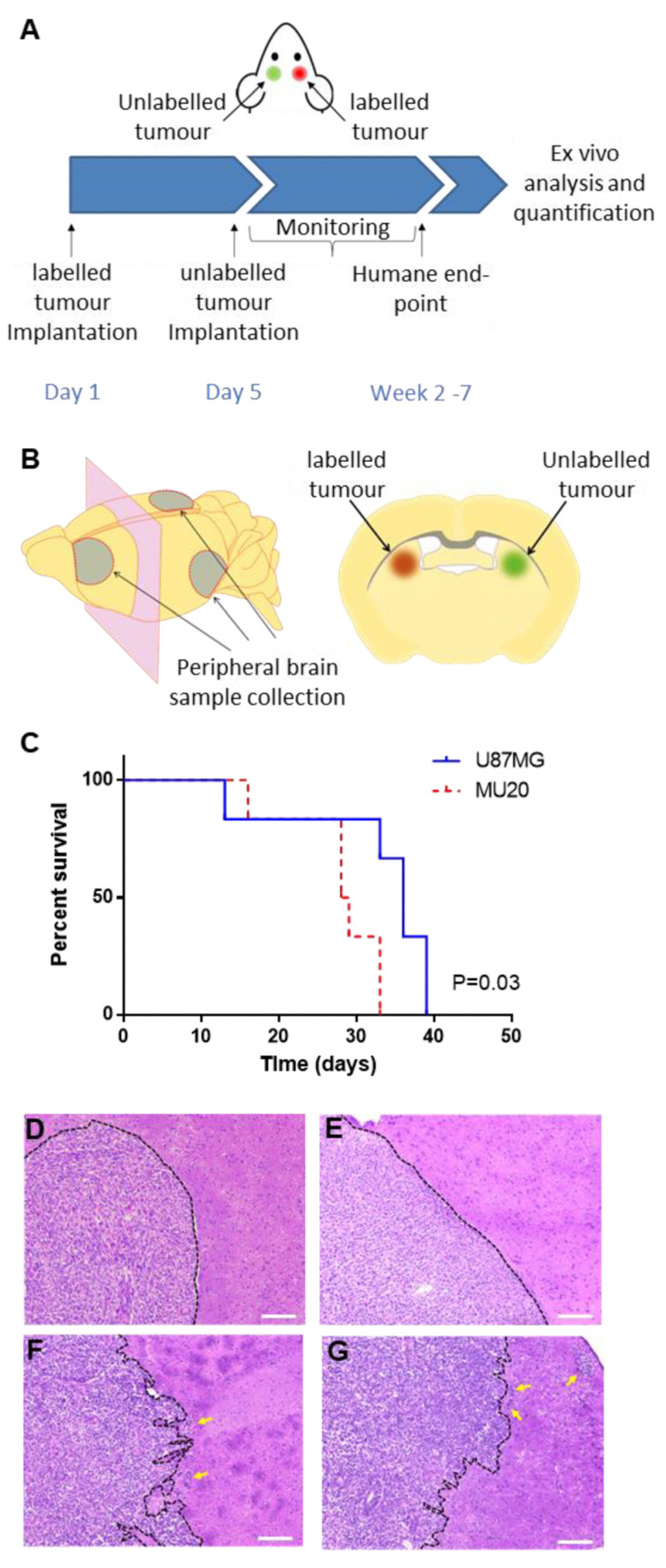
Orthotopic contralateral mouse implantation of U87MG and MU20 GBM cells displayed similar survival, with different patterns of tumour invasion. (**A**) Contralateral tumours were implanted on day 1 and day 5 using 5 × 10^4^ labelled or unlabelled tumour cells. The mice were monitored and euthanised at a humane end-point. (**B**) Brains were harvested for the collection of peripheral brain tissue samples (**left panel**) and contralateral tumours (**right panel**). (**C**) Kaplan–Meier curve of the overall survival rate for the mice bearing MU20 and U87MG contralateral implantations (n = 6). Microscopic images of the H plus E analysis of the labelled tumours for the U87MG-Fluc (**D**,**E**) and MU20-Fluc (**F**,**G**) cells (scale bar = 200 µm). The yellow arrows indicate the areas of tumour invasion in the MU20-Fluc xenografts.

**Figure 3 cells-13-00192-f003:**
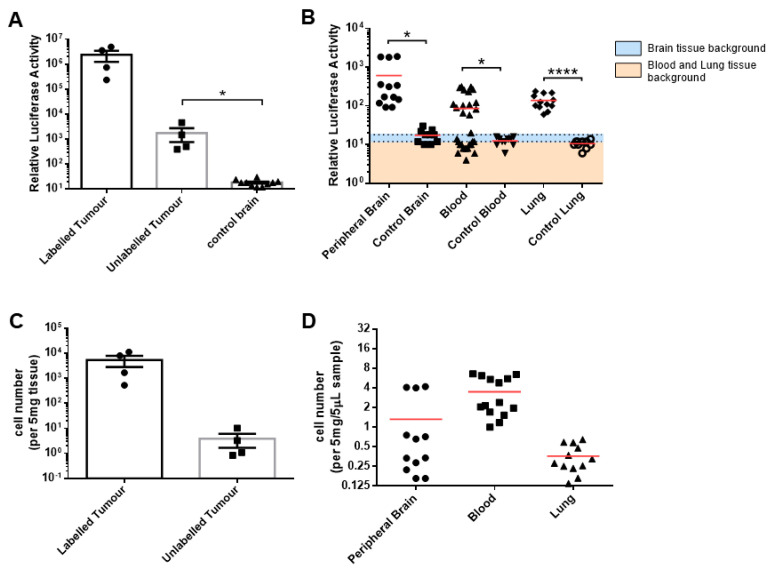
Systemic dissemination of the U87MG-Fluc cells in the brain, blood and lung samples. (**A**,**B**) We homogenised 5 mg of explant tissue or 5 µL of blood samples and lysed them for analysis of the relative luciferase activity (R.L.U), and we compared them to the peripheral and control tissues. The luciferase background was 18 R.L.U for the brain tissue and 12 R.L.U for the blood and lung tissues. (**C**,**D**) The cell number was quantified using titration curves calculated in vitro (Figure 1) for each tissue source. The data shown are the means of triplicates ± SEMs, with n = 4 (**A**,**C**) and the means of three independent tissue samples or eight independent blood samples per mouse, with n = 4 (**B**,**D**). For the control tissue and blood samples, n = 10 independent samples. * *p* < 0.05 and **** *p* < 0.0001; unpaired student’s *t* test.

**Figure 4 cells-13-00192-f004:**
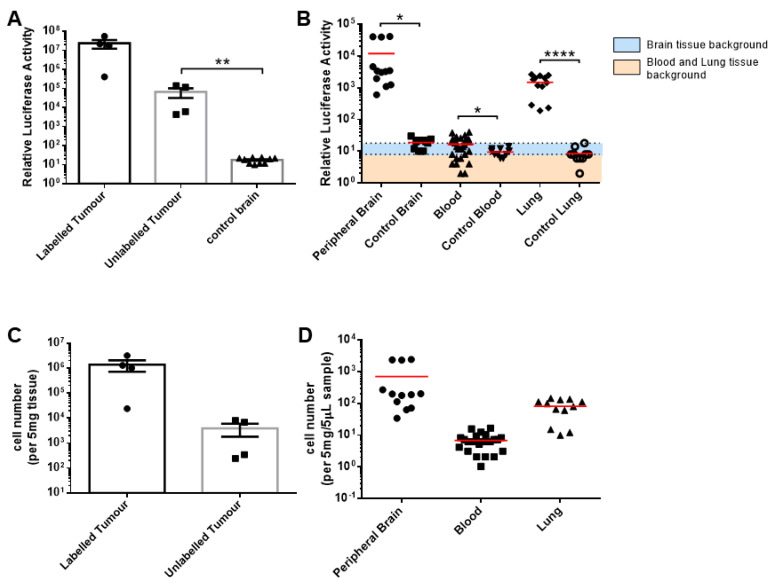
Systemic dissemination of the MU20-Fluc cells in the brain, blood and lung samples. (**A**,**B**) We homogenised 5 mg of explant tissue or 5 µL of blood samples and lysed them for analysis of the relative luciferase activity (R.L.U) and compared them to the peripheral and control tissues. The luciferase background was 18 R.L.U for the brain tissue and 8 R.L.U for the blood and lung tissue samples. (**C**,**D**) The cell number was quantified using the titration curves calculated in Appendix A Appendix A for each tissue source. The data shown are the means of triplicates ± SEMs, with n = 4 (**A**,**C**) and the means of three independent tissue samples or eight independent blood samples per mouse, with n = 4 (**B**,**D**).For the control tissue and blood, n = 10 independent samples. * *p* < 0.05; ** *p* < 0.01; and **** *p* < 0.0001; unpaired student’s *t* test.

**Figure 5 cells-13-00192-f005:**
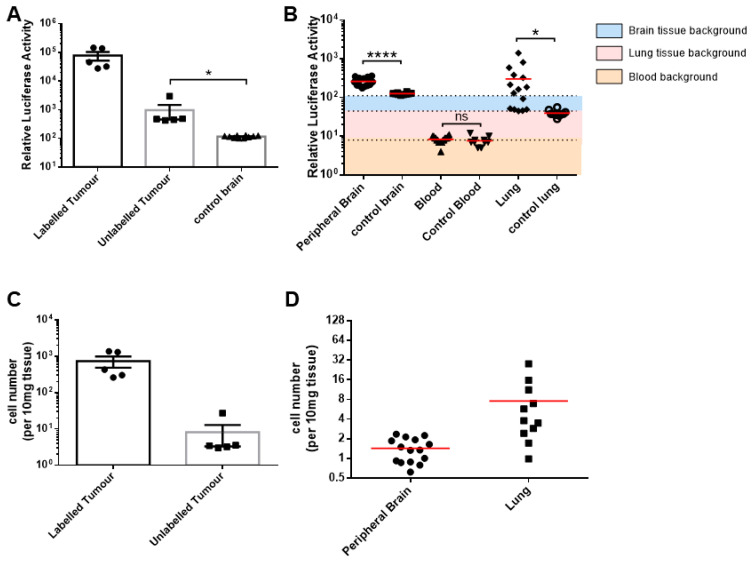
Systemic dissemination of the U87MG-Fluc cells in the brain and lung tissues 15 days after implantation. (**A**,**B**) We homogenised 10 mg of explant tissue or 5 µL of blood samples and lysed them for analysis of the relative luciferase activity (R.L.U) and compared them to the peripheral and control tissues. The luciferase background was 115 R.L.U for the brain tissue, 45 R.L.U for the lung tissue and 8 R.L.U for the blood samples. (**C**,**D**) The cell number was quantified using the titration curves calculated in vitro (Appendix A Appendix A) for each tissue source. The data shown are means of triplicates ± SEMs, with n = 5 (**A**,**C**) and the means of three independent tissue or blood samples per mouse, with n = 5 (**B**,**D**). For the control tissue and blood samples, n = 10 independent samples. * *p* < 0.05 and **** *p* < 0.0001; unpaired student’s *t* test.

## Data Availability

The data presented in this study are available on request from the corresponding author. The data are not publicly available because they are not currently in a data repository. The data are not currently in a data repository. The authors have full access to all the data in the study and take responsibility for the integrity of the data and the accuracy of the data analysis.

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
