# Peer review of "A New Systemic Disease Mouse Model for Glioblastoma Capable of Single-Tumour-Cell Detection"

_cells, 2024, doi:10.3390/cells13020192_

Round 1
Reviewer 1 Report
Comments and Suggestions for Authors
Glioblastoma is a lethal malignant brain tumor, and tumor detection will be pretty helpful to start the treatment earlier for recurrent glioblastoma. If we can detect a cell before it spreads throughout the brain, it will significantly help prolong their lives.
The experiment design is suitable to detect tumor cell detection using luciferase-labeled glioblastoma cells, and the results are precise and support their conclusion. At this point, they used mouse models, but if it can be extended to humans later, it will be a great technique to assess the glioblastoma progression of patients.
I recommend publishing this manuscript in Cells after these minor revisions.
1) There is an inconsistency when the author mentions luciferase-modified glioblastoma cells: -Fluc, -fluc, and F-Luc—for example, Fig. 1. U-87MG-F-Luc and U87MG-Flc in the same line.
2) A zero before decimals should either be consistently added or not. The manuscript has a P value. *P<0.05 and (blood, P=.04; lung tissue, P<.001). You cannot have both styles in one manuscript. My recommendation is with leading zero. Both 0.05 and .05 mean the same thing, but the leading zero can help reduce error when reading the numbers.
Author Response
Reviewer 1, Comment 1: There is an inconsistency when the author mentions luciferase-modified glioblastoma cells: -Fluc, -fluc, and F-Luc—for example, Fig. 1. U-87MG-F-Luc and U87MG-Flc in the same line.
Our comment addressing reviewer 1 comment 1 (R1.1): We have made the recommended changes in the manuscript. Firefly luciferase has now been consistently shortened to Fluc throughout the main and supplementary materials files. Likewise, Gaussian luciferase has been shortened to Gluc throughout the main and supplementary materials files.
Reviewer 1, Comment 2: A zero before decimals should either be consistently added or not. The manuscript has a P value. *P<0.05 and (blood, P=.04; lung tissue, P<.001). You cannot have both styles in one manuscript. My recommendation is with leading zero. Both 0.05 and .05 mean the same thing, but the leading zero can help reduce error when reading the numbers.
Our comment addressing R1.2: We have made the recommended changes in the manuscript. All P values have been changed to have a zero before the decimal point throughout the main text file. This was not required in the supplementary file.
Reviewer 2 Report
Comments and Suggestions for Authors
In this study Ware et al. have developed mouse model for glioblastoma capable of single tumour Cell Detection. However, I have some complaints.
1. In vivo models of luciferase-labeled glioma, examining the recurrence and metastasis of tumor cells, have long been developed (references: 10.2310/7290.2012.00029; 10.3791/53287; 10.7150/jca.15564; 10.1038/s41598-020-67411-w; 10.1186/s12967-014-0345-4). However, the authors do not cite any of these studies.
2. I don't understand the point of this paper. If the main point is that the authors have developed a model which enables detection of single cells and circulating tumor cells (CTCs) then this should be clearly emphasized. This paper should not extensively describe the pathogenesis and the failure of therapy but rather explain the shortcomings and advantages of the developed animal model compared to existing ones.
3. The authors should address the controversy surrounding the use of luciferase. Some studies report a robust immune response against luciferase (references: 10.7150/jca.15564; 10.1038/s41598-020-67411-w), while others deny such activation (reference: 10.1186/s12967-014-0345-4). Given this, the utility of luciferase seems limited to immunocompetent animals. The authors, who used NOD SCID mice in this study, should discuss the drawbacks associated with the use of immunocompromised animals in their experimental design.
4. The number of animals used in the study is limited. It's crucial for the authors to acknowledge and discuss this limitation.
Comments on the Quality of English LanguageNone.
Author Response
Reviewer 2, Comment 1: In vivo models of luciferase-labeled glioma, examining the recurrence and metastasis of tumor cells, have long been developed (references: 10.2310/7290.2012.00029; 10.3791/53287; 10.7150/jca.15564; 10.1038/s41598-020-67411-w; 10.1186/s12967-014-0345-4). However, the authors do not cite any of these studies.
Our comment addressing R2.1: Yes, we agree with the reviewer that these glioma-based references should be included. We have added a sentence in the introduction and cited the 4 papers suggested by the reviewer.
Reviewer 2, Comment 2: I don't understand the point of this paper. If the main point is that the authors have developed a model which enables detection of single cells and circulating tumor cells (CTCs) then this should be clearly emphasized. This paper should not extensively describe the pathogenesis and the failure of therapy but rather explain the shortcomings and advantages of the developed animal model compared to existing ones.
Our comment addressing R2.2: We agree with the reviewer that we should strengthen our emphasis of our key findings/points of this study. To reduce possible confusion, we have further emphasized the key points of this study in the discussion and conclusion. However, we disagree with the reviewer regarding their comments on reducing the description of glioblastoma pathogenesis and failure of therapy. One of our key objectives of this study was to establish a model that could eventually be used for therapeutic evaluation of novel or repurposed agents and thus we needed to introduce the fact that current mouse models are often not good models for testing potential therapeutic agents leading to pre-clinical success but clinical failure. This is therefore an important section we need to include in our manuscript.
Reviewer 2, Comment 3: The authors should address the controversy surrounding the use of luciferase. Some studies report a robust immune response against luciferase (references: 10.7150/jca.15564; 10.1038/s41598-020-67411-w), while others deny such activation (reference: 10.1186/s12967-014-0345-4). Given this, the utility of luciferase seems limited to immunocompetent animals. The authors, who used NOD SCID mice in this study, should discuss the drawbacks associated with the use of immunocompromised animals in their experimental design.
Our comment addressing R2.3: The reviewer is correct that in some cases luciferase may potentially elicit an immune response. Although in one of the 2 papers referenced by the reviewer, secreted luciferase models were used which is more likely to elicit an immune response. In the other paper referenced, the authors state that “While our study does not specifically elucidate the source of this immune response, we postulate that the presence of luciferase itself could be responsible”. Thus, although a possibility for an internalized protein to elicit an immune response, they did not prove that this enhanced immunity was specifically due to the luciferase protein specifically. Nonetheless, the reviewer is correct in suggesting that luciferase-based models in immune-competent mice should be used with caution and careful consideration. Our model used human glioblastoma cell lines in the immunocompromised mouse strain NOD SCID. We agree with the reviewer that we need to discuss the limitations of this model due to the inability of our model to evaluate anti-tumour immune effector function. We have added a sentence “However, one limitation of our model is the inability to evaluate any anti-tumour immune response promoted by potential therapeutics that elicit these immune-based responses.”to the discussion highlighting this limitation.
Reviewer 2, Comment 4: The number of animals used in the study is limited. It's crucial for the authors to acknowledge and discuss this limitation.
Our comment addressing R2.4: We acknowledge that our animal numbers are low, but we were still able to achieve our initial goals and show important and significant findings with both our in vivo and ex vivo data. Nonetheless, we have added the sentence “In addition, despite obtaining clear, significant outcomes in our animal model, greater number of mice will need to be used to validate other similar models in glioblastoma and other cancer settings.” To our discussion.
Reviewer 3 Report
Comments and Suggestions for Authors
In this study, the authors have established an innovative mouse model for glioblastoma that identified glioblastoma as a systemic brain disease, with individual tumor cells infiltrating all regions of the supratentorial brain. The continued exploration and application of this mouse model have the potential to enhance understanding of glioblastoma progression, suggesting novel research approaches for finding predictive biomarkers and targeted therapies to prevent recurrence.
The article is well-written, but additional explanations are needed, particularly in the Materials and Methods section.
In the Animal Experiments section, please add the end-point of the experiment.
Please provide a more detailed description of the experimental design. I suggest highlighting and explaining, right there, why the non-labeled contralateral tumor was implanted four days later compared to the implantation of firefly-labeled cells.
Furthermore, the Ex vivo Luciferase Analysis section also requires a more detailed description. Additionally, please mention that, alongside Frozen Tissue, blood was also used.
Please increase the font size and improve the resolution of Figure 2A. In 2D-2G, please indicate labeled tumors.
Considering that the study focuses on establishing a new glioblastoma model, it is essential to comprehensively describe the model establishment and signal detection.
Author Response
Reviewer 3, Comment 1: In the Animal Experiments section, please add the end-point of the experiment.
Our comment addressing R3.1: We agree that the endpoint of the mouse experiment should be included in the methods section. We have added “(based on daily monitoring and condition of the mice – ranging from 13 - 39 days after implantation of cells)” to the animal experiments section of the methods (section 2.3).
Reviewer 3, Comment 2: Please provide a more detailed description of the experimental design. I suggest highlighting and explaining, right there, why the non-labelled contralateral tumor was implanted four days later compared to the implantation of firefly-labelled cells.
Our comment addressing R3.2: We have provided a more detailed description of the animal experimentation procedure in the methods section (see revised manuscript). In addition, we have provided a description of why non-labelled contralateral tumors were implanted also in the same methods sub-section.
Reviewer 3, Comment 3: the Ex vivo Luciferase Analysis section also requires a more detailed description.
Our comment addressing R3.3: We have provided a more detailed description of the ex vivo luciferase analysis procedure in the methods section (see revised manuscript).
Reviewer 3, Comment 4: Additionally, please mention that, alongside Frozen Tissue, blood was also used.
Our comment addressing R3.4: Yes, blood collection was also added to the methods section – Animal experiments.
Reviewer 3, Comment 5: Please increase the font size and improve the resolution of Figure 2A.
Our comment addressing R3.5: Figure 2 has been improved including enlarging the font.
Reviewer 3, Comment 6: In 2D-2G, please indicate labelled tumors.
Our comment addressing R3.6: As outlined in the figure legend for Figure 2, all H+E images indicate labelled tumours of either U87Mg-Fluc or MU20-Fluc.
Reviewer 4 Report
Comments and Suggestions for Authors
The study 'A New Systemic Disease Mouse Model for Glioblastoma Capable of Single Tumour Cell Detection' by Ware et al. describes the development of a systemic disease model using mice implanted with glioblastoma cells. This allows for the investigation of recurrence biology and therapeutic efficacy. However, there are some significant concerns about this study:
1. It is unclear why the authors inoculated unlabeled cells in the contralateral hemisphere as recipient cells. This approach is not acceptable if the authors wanted to claim that glioblastoma is a systemic disease by artificial intervention.
2. The reproducibility of results is questionable. The authors claim single-cell level detection at peripheral brain cuts and that this approach is quantifiable for evaluating therapy efficacy. However, they did not provide adequate evidence to support this claim.
3. The detection of glioblastoma cells in the lung sections is an intriguing result that needs further explanation.
4. To prove the value of the developed models, treatments with at least one anti-glioblastoma agent mentioned in the introduction should be performed.
There are also some minor concerns:
1. The figures are of poor quality and unreadable due to the font being too small.
2. The immunohistochemical analysis seems to be a simple HE staining of brain tissue sections.
3. The experiment with adding glioblastoma cells ex vivo into the mouse brain seems pointless.
4. The study did not specify the number of mice or groups used.
Overall, there are serious objections to this study.
Author Response
Reviewer 4, Comment 1: It is unclear why the authors inoculated unlabelled cells in the contralateral hemisphere as recipient cells. This approach is not acceptable if the authors wanted to claim that glioblastoma is a systemic disease by artificial intervention.
Our comment addressing R4.1: Our model using a contralateral system has been performed elsewhere and is now a well-established model for detecting self-seeding of tumour cells. We have included a brief description in the methods section and results section indicating that this contralateral model (in other cancer settings) has been previously used to evaluate tumour cell infiltration/invasion and evaluate therapeutic efficacy. We also reference and briefly discuss a self-seeding glioblastoma paper in the discussion of our original manuscript which further supports the use of this model here in this manuscript. Thus, we do not agree with the reviewer that this is unacceptable as other models have used this similar method.
Reviewer 4, Comment 2: The reproducibility of results is questionable. The authors claim single-cell level detection at peripheral brain cuts and that this approach is quantifiable for evaluating therapy efficacy. However, they did not provide adequate evidence to support this claim.
Our comment addressing R4.2: We are unsure how the reviewer has formed this opinion that our data is not reproducible. We have performed two models that are reproducible with appropriate experimental controls within and across each experiment. The first evaluates the detection of labelled glioblastoma cells within mouse tissue, (“spiking” ex vivo experiment), while the second evaluates detection of labelled glioblastoma cells after orthotopic injection of cells (in vivo experiments). We indeed have clear evidence presented in both our main figures and supplementary data showing clear evidence. The reviewer is thus making a very serious accusation and question our scientific integrity by ignoring data evidences.
Admittingly, we did not perform an assessment of therapeutic efficacy using our model to determine if said agent could block glioblastoma cell infiltration as measured using our highly-sensitive luciferase-based in vivo model. Thus, we have changed our sentence “The ability to measure tumour infiltration in vivo with use of this model will provide a significant advancement over existing preclinical models that have failed to predict therapeutic efficacy in the clinic.” to “The ability to measure tumour infiltration in vivo with use of this model may provide a significant advancement over existing preclinical models that have failed to predict therapeutic efficacy in the clinic.”
We have also added the sentence: “This is critical as our model could offer an improved in vivo method to test the efficacy of novel and repurposed therapeutics in a reccurent setting.” to further emphasize our point that our model “could” provide improved pre-clinical testing but still requires therapeutic evaluation before making stronger claims.
Reviewer 4, Comment 3: The detection of glioblastoma cells in the lung sections is an intriguing result that needs further explanation.
Our comment addressing R4.3: We did indeed identify glioblastoma cells that had moved from the brain into the lung in our mouse model. To the best of our knowledge, tumour cells of glioblastoma origin have never been identified in the lung of patients clinically. However, we also do not know if the evaluation of lung micro-metastasis from glioblastoma has ever been performed and thus perhaps some glioblastoma patients do indeed have small lung metastasis that may be potentially identified if examined or may be undetectable when scanned. We speculate regarding this notion with the sentence “Although lung metastasis has not been reported from glioblastoma patients, we cannot rule out that micro-metastasis may occur but at such low cell number that are undetectable or that lung screening is simply not performed in the glioblastoma setting.” added to our discussion.
Reviewer 4, Comment 4: To prove the value of the developed models, treatments with at least one anti-glioblastoma agent mentioned in the introduction should be performed.
There are also some minor concerns:
Our comment addressing R4.4: The value of our model is for single cell detection and re-capturing GBM as a systemic disease. As such it can evaluate treatment efficacy to be better mimic patients. That’s next step. Indeed, we have follow up work using genetic approach treating GBM using this model to discover new mechanism specific for GBM and as such establishing new strategies for therapeutic development. However this is outside the scope of this manuscript.
Reviewer 4, Comment 5: The figures are of poor quality and unreadable due to the font being too small.
Our comment addressing R4.5: The figures have been modified.
Reviewer 4, Comment 6: The immunohistochemical analysis seems to be a simple HE staining of brain tissue sections.
Our comment addressing R4.6: Yes, the reviewer is correct that the mouse tumour tissue analysis is H+E. We apologize for this error and have removed immunohistochemical analysis from the Figure legend and replaced it with the more accurate H+E analysis.
Reviewer 4, Comment 7: The experiment with adding glioblastoma cells ex vivo into the mouse brain seems pointless.
Our comment addressing R4.7: We do not agree with the reviewer regarding our “spiking” experiment. In fact, we believe this experiment was essential to our overall study. Our initial aim was to determine if we could detect labelled glioblastoma cells within mouse tissue including the mouse brain thus performed our spiking experiment ex vivo (i.e.: to determine if the mouse tissue caused potential interference to the luciferase signal that was not seen with solitary labelled cells from tissue culture). Importantly, we could detect luciferase activity in as low as one labelled glioblastoma cell. Based on the success of these essential ex vivo experiments we only then moved on to perform our in vivo mouse experiments to identify glioblastoma infiltration and invasion into other parts of the brain and into circulation. Thus, we believe these experiments were essential to the work up of our key findings and not pointless. Let’s remember that the quality of any results is only as good as the controls.
Reviewer 4, Comment 8: The study did not specify the number of mice or groups used.
Our comment addressing R4.8: We have included the number of mice used into each relevant figure legend (i.e.: n=6).
Round 2
Reviewer 2 Report
Comments and Suggestions for Authors
The authors are not investigating a cure for gliomas and should refrain from providing extensive details on pathogenesis and therapy. Instead, the authors introduce a mouse model for glioblastoma, offering a potential tool for disease and therapy testing. It is essential for them to explicitly specify previously researched models similar to theirs and articulate the advantages of their model.
Comments on the Quality of English Language
The authors are not investigating a cure for gliomas and should refrain from providing extensive details on pathogenesis and therapy. Instead, the authors introduce a mouse model for glioblastoma, offering a potential tool for disease and therapy testing. It is essential for them to explicitly specify previously researched models similar to theirs and articulate the advantages of their model.
Author Response
Our introduction begins with highlighting the unusual zero success rate of over 1000 clinical trials on GBM while efficacy were confirmed in pre-clinical animal models (paragraph 1).
We then detailed the failure of three arguably most hopeful targeted therapy types in trials while succeeded in mouse models “measuring the growth rate of the primary tumour in subcutaneous and orthotopic intracranial glioblastoma implants” (manuscript text) (paragraph 2-4).
Then we analyzed the reasons of this discourse in paragraph 5, explicitly pointing out “Current preclinical models used to predict therapeutic efficacy most often report the response of the primary tumour” while in real world the response to infiltrated tumor cells is needed. As such a different model mimicking GBM as a systemic disease is urgently needed.
Consequently, we believe our introduction has addressed reviewer’s comments in substance. though we may have not done so in a way the reviewer would like.
We would like to keep our way of writing.
Reviewer 4 Report
Comments and Suggestions for Authors
The authors did not adequately address my concerns.
The quality of Figure 2 is still poor. I cannot identify the Fluc cells indicated by arrows in the HE-stained sections. Additionally, the authors did not convince me that their method of glioblastoma cell quantification by luciferase luminescence correlation with the number of cells is accurate.
Author Response
We disagree with the reviewer’s remark about the clarity and quality of the figures. However they do require to be viewed on appropriate screen. By the way, the arrows are simply pointing to tumor cells which are F-luc labelled. Moreover, Cells is an open, electronic publication journal. It’s effortless to enlarge figures and text on screen to have clear view.
In Figure 1 we presented data in vitro showing 1, varied background luciferase for different tissues; 2, different sensitivity in different tissues; 3, clear single cell sensitivity in all tissues; and 4, cell number accuracy in mathematic formulations.
Finally, in vivo the data distributions (particularly being compared with control tissues) in Figure 3-5 clearly demonstrated single cell detection and measurement.
We are proud of our thoroughness in establishing the methodology of cancer cell number measurement.